# Extrafloral Nectary-Bearing Plants Recover Ant Association Benefits Faster and More Effectively after Frost-Fire Events Than Frost

**DOI:** 10.3390/plants12203592

**Published:** 2023-10-17

**Authors:** Gabriela Fraga Porto, José Henrique Pezzonia, Kleber Del-Claro

**Affiliations:** 1Programa de Pós-Graduação em Biologia Vegetal, Instituto de Biologia, Universidade de Uberlândia, Uberlândia 38400-902, MG, Brazil; gabrielafraga@usp.br; 2Programa de Pós-Graduação em Entomologia, Faculdade de Filosofia, Ciências e Letras de Ribeirão Preto—FFCLRP, Universidade de São Paulo, Ribeirão Preto 14040-901, SP, Brazil; jh.pezzonia@usp.br; 3Laboratório de Ecologia Comportamental e de Interações, Instituto de Biologia, Universidade Federal de Uberlândia, Uberlândia 38400-902, MG, Brazil

**Keywords:** mutualism, ant–plant interactions, Cerrado, tropical savanna, myrmecophily

## Abstract

The Cerrado confronts threats such as fire and frost due to natural or human-induced factors. These disturbances trigger attribute changes that impact biodiversity. Given escalating climate extremes, understanding the effects of these phenomena on ecological relationships is crucial for biodiversity conservation. To understand how fire and frost affect interactions and influence biological communities in the Cerrado, our study aimed to comprehend the effects of these two disturbances on extrafloral nectar (EFN)-bearing plants (*Ouratea spectabilis*, Ochnaceae) and their interactions. Our main hypothesis was that plants affected by fire would grow again more quickly than those affected only by frost due to the better adaptation of Cerrado flora to fire. The results showed that fire accelerated the regrowth of *O. spectabilis*. Regrowth in plants with EFNs attracted ants that proved to be efficient in removing herbivores, significantly reducing foliar herbivory rates in this species, when compared to the species without EFNs, or when ant access was prevented through experimental manipulation. Post-disturbance ant and herbivore populations were low, with frost leading to greater reductions. Ant richness and diversity are higher where frost precedes fire, suggesting that fire restores Cerrado ecological interactions better than frost, with less impact on plants, ants, and herbivores.

## 1. Introduction

Since the last century, we have known that plant–animal interactions are ubiquitous and extensively present in terrestrial environments. These interactions are mainly responsible for the structure of trophic chains [1,2]. Recently, some authors [3,4] examined mutualistic networks between plants and animals in various biomes worldwide. Their study revealed that animal–plant interactions create intricate and interconnected networks that significantly influence community diversity and stability. These mutualistic interactions, such as pollination, seed dispersion, and protective interactions, are recognized as crucial factors driving the evolution and preservation of biodiversity [5]. Indeed, Bronstein (2021) [6] suggested that “it is increasingly apparent that mutualisms are central components of ecological communities, generating linkages among species in ways that lead to rich and persistent assemblages”.

The protective mutualism between ants and plants has been pointed out as one of the most widespread and pervasive biotic interactions in the Neotropics [7], serving as a good model to improve our understanding of the impacts of climate change, fire, and other anthropogenic actions on natural communities [8,9,10]. In these mutualistic relationships, the primary resource that plants offer to attract protective ants is the nectar produced by their extrafloral nectaries (EFNs) [11]. EFNs are specialized structures that secrete nectar but are not involved in pollination. Instead, they are typically found on various above-ground plant parts, including leaves, stems, stipules, and flower buds. EFNs produce a liquid consisting of carbohydrate-rich compounds, together with small quantities of other organic compounds like amino acids and lipids [12,13]. EFNs act as induced defense mechanisms for plants [14,15]. Due to ant attendance, plants experience reduced herbivory and increased fruit set production [16,17]. On the other hand, by feeding on the nectar, ants significantly improve the health and survival of their colonies [18,19,20,21]. EFNs are present in at least 3941 species distributed across 745 genera and 108 families [22]. EFN-bearing plants play a significant role in the overall vegetation composition, particularly in tropical rainforests. For instance, a single hectare of Australian rainforest can harbor nearly 30 different species of EFN-bearing plants [23]. These plants are also prevalent in other ecosystems such as Mexico’s seasonal forests [24], arid environments in Argentina [20], and even the Brazilian Cerrado [25].

The Cerrado is a highly diverse biome that originally covered over 25% of the Brazilian territory; however, today, less than 8% of its area remains preserved [26]. Fire is a crucial evolutionary force in this biome and plays a fundamental role in maintaining biomass gradients and vegetation formations [27,28]. While fires can occur naturally due to atmospheric electrical discharges (e.g., lightning) or be intentionally set for ecosystem restoration and pasture management [29,30], the greatest impact of fire comes from human activities [28,31]. Over recent decades, the Cerrado has experienced various anthropogenic disturbances such as habitat fragmentation, extraction of timber and non-timber products, livestock farming, invasion of exotic species, alterations in burning regimes, and carbon cycle imbalance [28,32,33]. Human development has led to changes in terrestrial landscapes and associated biological communities worldwide [34], contributing to climate change and an increase in extreme weather events [35].

Despite fire being one of the most important factors determining the community structure in the Cerrado, this biome is also susceptible to frost, which can be considered relatively frequent in Southern Hemisphere savannas [36]. A study conducted in South Australia demonstrated that despite the global trend of increasing temperatures, the duration of the frost season has also increased [37]. Although the duration of winter has been noted to decrease due to global warming, paradoxically, global warming can increase the risk of frost events [38]. Researchers have faced a paradoxical scenario between plant growth and climate change, with evidence confirming that increased warming raises the risk of frost damage to plants [39]. Hot climates can induce premature growth in plants, which can be interrupted by frost events. These fluctuations between hot and cold weather due to extreme climatic conditions can act as “surprise agents”, posing a risk to vegetation [39,40]. Frosts can cause changes in plant phenology [41,42] and influence plant interactions with other organisms, such as insects [43]. However, most studies examining organism responses to frosts are limited to temperate regions, with very few studies focused on tropical biomes. Among those few studies, the emphasis has been primarily on vegetation responses to this event [44,45,46]. This highlights the limited knowledge of the impact of this physical stressor on animals and biotic interactions in tropical regions.

In the Cerrado, at least 25% of woody species in the tree flora have extrafloral nectaries [47]. In plants subjected to fire, newly regrowing leaves tend to have better nutritional quality and are often accompanied by the secretion of nectar from EFNs [48,49,50]. Rapid regrowth and foliar nutrition can benefit many herbivore species [51]. However, not all herbivores are successful in exploiting their host due to the defense mechanisms employed by the plants. Continuous fire events can lead to changes in organism interactions [27,52,53]. These variations include the outcomes of mutualistic interactions between ants and plants with EFNs [14,17,49,54]. Similarly, frosts can affect both plant and animal species [43,55], potentially leading to variations in interactions between ants and plants with extrafloral nectaries, an effect that is still unknown in the Brazilian savanna.

To understand how fire and frost affect interactions and influence biological communities in the Cerrado, this study aimed to comprehend the effects of these two disturbances on EFN-bearing plants and their interactions. In late July 2021, two strong frosts hit a reserve in the Cerrado region of Uberlândia, Brazil. The frosts resulted in the cold-induced burning and shedding of leaves from almost all plants, including herbs, shrubs, and trees. In early September, 2 months after the frost events, the area was struck by a massive 3-day fire that engulfed almost the entire reserve, leaving only a small area of less than 10% untouched by the fire. These events provided the opportunity to simultaneously evaluate the effects of frost and fire on plants and their biotic interactions in the same area. Thus, *Ouratea spectabilis* (Mart.) Engl., an abundant EFN-bearing tree in the area for which interaction data from the same area are already available in the literature [56], was chosen as a model. The following topics were addressed after frosts and after frosts and fire: (1) the regrowth time of *O. spectabilis*; (2) the variation in abundance and diversity of associated ants and herbivores; (3) and the outcomes of biotic interactions (EFN-bearing plant-ant-herbivore), measured by leaf area loss.

We proposed three hypotheses to be tested directly under field conditions. The first hypothesis suggests that plants affected by fire would grow again more quickly than those affected only by frost due to the better adaptation of Cerrado flora to fire. The central question raised by this hypothesis is whether the recovery periods of plants after frost and after frost and fire differ. Previous studies have demonstrated the resilience of ants and plants with EFNs to fire in the Cerrado [14,57,58]. Hence, our second hypothesis posits that in the area affected by frosts and subsequent fires, if this area indeed grows again more quickly, the protective behavior provided by ants would be more efficient compared to the area affected only by frosts. The main questions associated with this hypothesis are: (i) How did these disturbances affect the ant and herbivore fauna associated with *O. spectabilis* differently? (ii) Did the protective behavior of ants vary between species with and without EFNs under the influence of frosts and frosts followed by fire? (iii) Do regrowing plants with EFNs experience less damage from herbivores compared to plants without EFNs due to the biotic protection provided by ants? Our third hypothesis suggests that plants with EFNs, compared to plants without these glands, will exhibit lower rates of herbivory due to the biotic protection provided by ants. The central question here is whether there is an association between the diversity and abundance of ants and the rates of herbivory on plants after fire, as well as after frosts and fire.

## 2. Results

### 2.1. Regrowth Time

The regrowth of *O. spectabilis* individuals analyzed under the effects of frost and fire occurred at a faster rate than the individuals affected only by frost (Figure 1B). One month after the fire, the plants that experienced the effects of fire already showed regrowth intensity of up to 25%, and 2 months later, they exhibited regrowth intensity of up to 50%. Meanwhile, the plants affected only by frost required 3 months to reach a regrowth intensity of up to 50% (Figure 1A). Despite this initial difference in the regrowth period, in both areas, post-disturbance recovery remained constant, with individuals from both areas achieving higher regrowth intensity during the same period, from February to April of the following year.

### 2.2. Herbivory in the Area under the Effects of Frosts and under the Effects of Frosts and Fire

The results showed that when analyzed independently, the factor (Species) was significant, meaning that it influenced the herbivory rate (GLM: χ^2^ = 49.4242; df = 1; *p* < 0.001). However, the factor (Area) was not significant (GLM: χ^2^ = 0.3356; df = 1; *p* = 0.5624), as well as the interaction between the factors (Species×Area) (GLM: χ^2^ = 0.0465; df = 1; *p* = 0.8293) (Figure 2). The average herbivory rates for species with and without EFNs in the frost-affected area were as follows: *O. spectabilis*, 1.22 ± 0.200, and *A. tomentosum*, 2.55 ± 0.188 (mean ± standard deviation). Similarly, in the area subjected to both frost and subsequent fire, the average herbivory rates were: *O. spectabilis*, 1.06 ± 0.203, and *A. tomentosum*, 2.48 ± 0.188 (mean ± standard deviation). Thus, the average herbivory rate was higher in the species without EFNs in the areas under the effects of both disturbances.

### 2.3. Effectiveness of Ant Defense

The herbivory results showed that there was no significant difference in the herbivory rate between the control group and the treatment group (ants absent) of *O. spectabilis* in the area under the effects of frosts (GLM: χ^2^ = 1.8167; *p* = 0.1777; control 1.10 ± 0.18; treatment 1.44 ± 0.17; mean ± standard deviation; Figure 3). However, for the area under the effects of frosts and fire, the results showed a significant difference between the control group and the treatment group (GLM: χ^2^ = 10.02; *p* = 0.001). Plants without ants experienced higher foliar herbivory (control 1.37 ± 0.21; treatment 2.35 ± 0.21; mean ± standard deviation; Figure 3).

### 2.4. Effectiveness of Ant Predatory Behavior

The results showed significant differences in ant aggressiveness between species with and without EFNs in the area under the effects of frosts and in the area under the effects of frosts and fire (χ^2^ = 68.2, df = 3, *p* < 0.01; Figure 4). None of the termites used in the removal experiment was preyed upon in the species without EFNs in both areas. In contrast, in the species with EFNs, the ants demonstrated efficiency in termite removal in both areas; only one termite was not preyed upon in the area under the effects of frosts and fire, and in 240 s of the experiment, over 50% of the termites had already been preyed upon in both locations. However, the mean survival probability time was slightly lower for the termites tested in the area under the effects of frosts compared to those in the area under the effects of frosts and fire.

### 2.5. Ant Diversity

The diversity profile estimated by the Hill number showed higher species richness and dominance of ant species in the area under the effects of frosts and fire for all calculated diversity indices (Figure 5).

### 2.6. Abundance of Ants and Herbivores

During the 3 months of observations and collections of ants and herbivores conducted weekly, 11 ant species were reported in the area under the effects of frosts and fire, with *Camponotus crassus* (Mayr, 1862) being the most abundant species in both locations. Of the reported herbivores, 13 were collected in the area under the effects of frosts, and 26 in the area under the effects of frosts and fire. The GLM revealed significant differences in the mean abundance of herbivores between the areas (GLM: χ^2^ = 0.799; df = 16; *p* = 0.05; under frosts 0.26 ± 0.30 and under frosts and fire 1.06 ± 0.23; mean ± standard deviation). However, the ANOVA did not show significant differences in the mean abundance of ants (under frosts 1.06 ± 0.40 and under frosts and fire 1.84 ± 0.40; mean ± standard deviation; Figure 6).

## 3. Discussion

Plants affected by fire grew again more quickly than plants affected only by frost, showing a better adaptation of the Cerrado’s flora to fire occurrence, corroborating our first hypothesis. In both areas, the regrowth in plants with EFNs attracted ants that proved to be efficient in removing herbivores, significantly reducing leaf herbivory rates in this species when compared to a species without EFNs or when ant access was prevented through experimental manipulation. Thus, our second hypothesis that in the area affected by frosts and subsequent fire, the protective behavior provided by ants would be more efficient compared to the area affected only by frosts, was also corroborated. Ant and herbivore populations were reduced during the disturbances, with frosts causing a more significant reduction in both herbivores and ant populations. Additionally, ant richness and diversity were higher in the area that experienced frosts and subsequent fire. These results, together with the significant reduction in leaf area loss in plants with EFNs in the area affected by frost and fire, confirm our third hypothesis. Thus, in the Cerrado, plants with EFNs tend to exhibit lower rates of herbivory due to the biotic protection provided by ants [7,11,17]. Our main results indicated that fire plays an important role in restoring ecological interactions in the Cerrado, impacting plants, ants, and herbivores less negatively than frost. However, fire obviously results in a clear reduction in the overall fauna.

### 3.1. Regrowth Time

The regrowth of individuals observed in the area affected by frosts and fire occurred more rapidly compared to the area that experienced only frost effects. This may be occurring because Cerrado plants are better adapted to fire [59,60], even though frost injuries share important characteristics with fire damage [61], especially concerning internal stem, branch, and leaf tissue damage [62]. These factors affect regrowth, growth, and plant persistence [61]. Furthermore, we suggest that intense fire may have removed the preceding dead leaves that were affected by frost, which could justify the accelerated regrowth in the area where fire occurred compared to the one that only experienced frost events. Hoffmann et al. (2018) [61] mention that frost events occur in savannas frequently and with enough severity to act as an environmental filter and evolutionary selective factor for some of the species most sensitive to this event. Trees in warmer climates may experience a higher risk of frost effects due to their lower resistance to lower temperatures [63,64,65]. However, a recent study suggests that individual plant species’ characteristics, such as bark properties and bud protection, may be two important strategies for the protection and persistence of savanna trees under extreme temperature conditions. *O. Spectabilis* was one of the species analyzed in this study and was considered frost-resistant, bearing buds with a high degree of protection against extreme temperatures [36]. However, it is still not well understood how plants can tolerate lower temperatures in these areas, and even less well known are the effects of low temperatures on plant interactions with other organisms in the Cerrado. Studies conducted in temperate regions indicate, for example, alterations in leaf morphology, nectar composition, and chemical defenses of plants post-frost [66,67]. Such changes in plant characteristics due to frost events can result in alterations in the dynamics of plant interactions with other organisms, which are still poorly understood, especially in the Brazilian savanna. Therefore, conducting studies that consider the effect of these disturbances on interactions is becoming increasingly relevant.

### 3.2. Herbivory Rates

Compared to the leaves of the species with EFNs, the leaves of the species without EFNs consistently experienced higher herbivory rates in both areas. This result corroborates with studies [21,47,68] that identify EFNs as an effective biotic defense against herbivore action. It is important to note that both species exhibited low herbivory rates, even considering the emergence of fresh tissues as a favorable scenario for herbivores during post-fire regrowth [48,49]. Low herbivory levels in post-fire regrowth plants have also been reported in other studies [69,70], sometimes associated with higher nutrient availability (e.g., nitrogen) that leads to changes in secondary metabolites, increasing the production of defense compounds in plants [71]. However, Del-Claro and Marquis (2015) [72] demonstrate that interactions between ants and EFN-bearing plants are resilient to fire’s effects, and the associated ant species appears to be more crucial than fire. Given that the ant species *Camponotus crassus* Mayr, 1862, an important predator in the Cerrado, was the most abundant in both areas, it is suggested that low herbivory is associated with biotic defense provided by ants, especially in the area that experienced subsequent fire action.

### 3.3. Effectiveness of Ant Defense

The results of the experiment investigating the role of EFNs as a biotic defense clearly demonstrate that ants were efficient in protecting the plants in the area affected by frosts and fire, corroborating studies that indicate the resilience of ant–plant interactions with EFNs to fire [72]. However, in the area affected by frosts, herbivory rates did not differ between the treatment where ants had no access to the plants and the control where ants continued to have free access to them. We suggest that this result may be associated with the low number of herbivores found in this area; other studies also point to a reduction in herbivores after frost events [43,55]. Studies show that frost can directly affect the insect fauna by disrupting their metabolism and causing cell death through the formation of ice crystals [73] and indirectly by killing foliage and reducing their food resource [55,74]. Only one study had previously assessed the effects of frost on the herbivore community in Cerrado vegetation. Lopes (1995) [75] reported a decline in nymphs and adult membracids on host plants 1 to 2 months after frosts. This lone study highlights the lack of knowledge regarding how this disturbance can directly and indirectly affect the insect fauna and, consequently, the ecological interactions that this class promotes with organisms at other trophic levels.

### 3.4. Effectiveness of Ant Predatory Behavior

The experiment analyzing the predatory behavior of ants supported the herbivory results, highlighting the efficiency of EFNs as a biotic defense and the active role of ants in this interaction. During the experiment, all termites were removed by ants on EFN-bearing plants in the area affected by frosts, while only one was not removed in the area affected by frosts and fire. In the species without EFNs, no termites were removed in both areas. The ant *C. crassus* was responsible for removing all termites in the area affected by frosts. The species *C. crassus* is dominant in the Cerrado and displays highly aggressive behavior towards herbivorous insects [56,76]. This species was predominant in that area, which may explain its greater ability to prey on termites during the analysis of ant predatory behavior. In a recent study, this same species was considered the fastest and most efficient in capturing termites on nectariferous plants, demonstrating its capacity to act as a biotic defense [77]. Calixto et al. (2021) [78] also investigated the predatory behavior of *C. crassus* and demonstrated its efficiency in termite removal from five sympatric plant species with EFNs. Studies have shown greater protective benefits for plants visited by a single dominant and aggressive ant species compared to plants visited by multiple ant species [72,79].

In our study, the richness and diversity of ants in the two areas differed, and the results indicated that species richness and diversity were higher in the area affected by frosts and fire. While no study had previously assessed the effects of frosts on ant abundance and diversity in the Cerrado, studies that have evaluated the effect of fire have pointed to ants as being resistant and resilient to this disturbance [57,58,80], demonstrating that these organisms may have adapted over time. This could explain the greater richness and diversity of ants in the area where frosts and subsequent fire occurred. Some studies indicate that low temperatures can negatively impact ants by affecting mortality rates, while high and moderate temperatures may allow ants better access to resources, leading to increased feeding and lower mortality rates [81,82,83]. In other words, temperature is a limiting factor for the survival and activity of these organisms. Therefore, considering the current context of climate change, if some species have more conserved climatic niches than others, as demonstrated by other studies [84,85], these lineages may fail to develop the necessary characteristics to adapt to drastic climate changes [86,87]. Therefore, it becomes evident that the increase in anthropogenic disturbances and extreme climatic events seen today can have significant implications for ecosystem functionality, as ants play a key role in ecosystem processes such as nutrient cycling, seed dispersal, and structuring invertebrate communities through predation and competition [88,89]. Given the importance of ants, including their protective role mediated by EFNs, variations in their abundance and diversity can influence the interactions they engage in with other organisms. In a recent study, for example, Yamawo et al. (2021) [90] found negative relationships between ant species richness and the strength of interactions involving aggressive ant species. The authors observed that the effectiveness of indirect ant defense was low in areas where ant species richness was high. This same pattern has been reported in other studies [91,92], where the authors suggest that resource, space, or time limitations may reduce the interaction capacity of some species. Moura et al. (2022) [93] also found negative associations between ant nest richness and the herbivory rate of EFNs’ bearing plants. These authors suggest that intensified competition due to high nest richness may distract visiting ants from the presence of herbivores on the plants. In this study, direct associations between local ant diversity and herbivory rates were not found, likely due to the low number of herbivores found in both areas. However, the greater efficiency in predation time in the area affected by frosts, where the ant *C. crassus* was predominant, may indicate that a dominant and aggressive ant alone in a mutualistic system between EFN-bearing plants and ants can lead to more efficient outcomes for the established mutualistic relationship. Nevertheless, little is still known about how disturbances like frosts can influence interactions, highlighting the importance of promoting further studies to assess the effect of this and other disturbances in the environment.

## 4. Methods

### 4.1. Study Site

The study was conducted in a Cerrado (Brazilian tropical savanna) area within the legal reserve area of the “Clube de Caça e Pesca Itororó (CCPIU)” in Uberlândia, State of Minas Gerais, southeastern Brazil (18°59′569″ S–48°18′351″ W) (127 ha). According to the Köppen classification, the climate in the region is of type AW, with two well-defined seasons: a dry season from May to September and a rainy season from October to April. The average annual temperature is 22 °C, and the average annual precipitation is 1500 mm [94]. The vegetation in the reserve includes different types such as open grassland or savanna with sparse or no trees (Campo limpo); grassland with scattered shrubs and small trees (Campo sujo); a savanna characterized by a combination of grasslands, scrublands, and areas with scattered trees and shrubs (Cerrado); a dense, more wooded or forested area (Cerradão); and “Veredas”, that is, palm swamp or wetland surrounded by Cerrado vegetation [57]. We worked in the Cerrado area that experienced a strong frost in July 2021 and in the Cerrado area that experienced frost and was burned in early September.

### 4.2. Study System

The family Ochnaceae is composed of 27 genera and 500 tropical species distributed across tropical and subtropical zones worldwide [95]. The genus *Ouratea* stands out as the most diverse in this family, with around 300 tropical species primarily found in South America and tropical Africa [96]. Among the numerous plant species occurring in the Cerrado, *Ouratea spectabilis* stands out as one of the most abundant and common species [97]. *O. spectabilis* possesses EFNs at the base of its stipules, which attract various ant species and promote interactions between ants and herbivores.

Popularly known as “peroba-do-campo”, *Aspidosperma tomentosum* Mart. is a species belonging to the Apocynaceae family. This species is characterized as semi-deciduous, heliophilous, or partially shade-tolerant and can be found in phytogeographic provinces of the Chaco and Cerrado in Bolivia, Brazil, and Paraguay [98]. The choice of this species for this study was motivated by its simultaneous regrowth with *O. spectabilis* and the absence of EFNs.

### 4.3. Leaf Regrowth Analysis

One week after the fire occurrence, in the last week of September for the leaf regrowth analysis, we randomly selected 15 individuals of *O. spectabilis* in the area affected by frosts and another 15 individuals in the area affected by frosts and fire. The individuals had similar sizes (3–3.5 m tall) and were separated by at least 10 m of distance. The plants were inspected three times a week for a period of 8 months during the morning. Within this time frame, data on the intensity of new leaf presence in individuals from each area were recorded. The methodology used for standardizing the records was proposed by Fournier (1974) [99], which has been adapted and previously utilized in other studies [100,101]. This system is based on a semi-quantitative evaluation method where scores are assigned to each phenological phase. Each individual is assigned a value from 0 to 4 to indicate new leaf presence, with 0 indicating absence, 1 indicating presence from 1 to 25%, 2 indicating presence from 26 to 50%, 3 indicating presence from 51 to 75%, and 4 indicating presence from 76 to 100% of leaves (e.g., 56).

### 4.4. Identification of Ant and Herbivore Guilds

Ants and herbivores were monitored for 3 months after the disturbance through visual observations and the aid of photographic devices. Collections and observations were conducted 3 times a week, with each session lasting 40 min for each individual, between 7 a.m. and 5 p.m. Whenever possible, one specimen of each ant and herbivore species was collected for later identification in the Laboratory of Behavioral Ecology and Interactions (LECI). Collections were carried out using containers filled with 70% alcohol, and all herbivores and ants were quantified and classified to the species level whenever possible. We identified each collected ant individual (according to Baccaro et al., 2015, taxonomic key) [102,103,104,105,106]. The data on ant and herbivore abundance were recorded in plants of *O. spectabilis*.

### 4.5. Analysis of Ant Defense Effectiveness in Plants with EFNs

To test whether regrowing plants with EFNs exhibit lower herbivory rates compared to regrowing plants without EFNs due to the presence of attractive resources for natural enemies of herbivores (ants), we randomly selected 15 individuals of *O. spectabilis* of the same size (3–3.5 m tall) and phenological state in the frost-affected area and another 15 individuals following the same criteria in the area affected by frost and fire. We used a species that does not have EFNs for comparison. Thus, 15 individuals of *Aspidosperma tomentosum* Mart. (Apocynaceae) were selected in the frost-affected area and the area affected by both frost and fire, following the same criteria as for *O. spectabilis*. Additionally, to evaluate whether the herbivory rate is influenced by the presence or absence of ants on plants with EFNs and by frost and fire events, we used marked *O. spectabilis* individuals in the areas affected by frost and frost with fire, and we prevented ant access. We applied a non-toxic resin (Formifuu^®^) to block ant access to one branch of each individual, while allowing ants to access other parts of the plant. The herbivory rate was measured using photographic images of scanned leaves analyzed in the ImageJ (ij153-win-java8). Herbivory levels were expressed as the percentage of leaf area lost, estimated by the formula: Herbivory = [lost area/total leaf area] × 100.

### 4.6. Analysis of the Effectiveness of Ant Predatory Behavior

To analyze the indirect effect of ants on herbivores and their behavior in plants with and without EFNs, we selected 15 individuals of *O. spectabilis* in the frost-affected area and another 15 individuals in the area affected by both frost and fire. Once again, we used *A. tomentosum* to represent the species without EFNs and without herbivorous trophobionts. For the aggression and herbivore removal test by visiting ants, the herbivores used were workers of *Nasutitermes* sp. termites, which are prey commonly used in experiments to analyze ant predatory behavior [77,78]. We used three live termites at a time per tested plant, placing each termite on leaves from three different branches. We used regular non-toxic white glue to prevent termite movement, being careful not to use an excessive amount to avoid hindering termite removal by ants. Two researchers were responsible for observing the interactions between ants and termites and recording the number of ants present on the plant during each experiment, the time it took for the ants to find the termite, and the behavior they exhibited (e.g., removed, expelled, or ignored). Each selected plant was analyzed for 20 min between 08:00 and 12:00 in the morning.

### 4.7. Statistical Analyses

To produce graphs displaying the regrowth time of the areas under different disturbances, we used a “plotrix” package [107] from the free software R version 4.1.2. All other statistical analyses and graphs were also performed using the same software.

To assess whether the herbivory rates differed among plant species with and without EFNs and among areas affected by frost and areas affected by both frost and fire, generalized linear models (GLM) with negative binomial distribution were employed to account for overdispersion. Assumptions of normality and homoscedasticity were examined using a DHARMa package in R [108], followed by post hoc tests with estimated marginal means using the “emmeans” package [109]. The analyzed factors were Species (with EFNs and without EFNs) and Area (frost-affected and frost- and fire-affected), together with their interaction. Differences were considered significant when *p* ≤ 0.05. To compare the efficiency of ant exclusion between plants with and without EFNs, a survival analysis with the Weibull distribution was performed using the “survival” package [110] for R. In this survival model, the termite exclusion events (a binary variable, with 1 = removal and 0 = no removal) and the time to exclusion were used as response variables, while the plants under frost-affected and frost- and fire-affected conditions were used as predictor variables. For this analysis, we used the Harrington and Fleming G-rho family (1982) [111] with the “survdiff” function from the “survival” package. Subsequently, paired comparisons were conducted using the log-rank test, and *p*-values were corrected using the Benjamini and Hochberg method in the “survminer” package [112]. To compare the diversity of ant species in the areas under different disturbances, we used the Hill diversity profile approach with the “vegan” package [113]. Hill numbers are a family of diversity indices that integrate species richness and species abundance across diversity orders [114]. Different Hill entropy values correspond to different diversity indices depending on the order q, where: q = 0 (equivalent to species richness), q = 1 (equivalent to diversity using the Shannon index), q = 2 (equivalent to species dominance using the Simpson index), and q = 3, 4, 5 (correspond to the weighting of rare species). We also conducted ANOVA and GLM with a negative binomial distribution to compare the mean abundance of ant and herbivore species in both areas, respectively.

## 5. Conclusions

Our study is the first to assess the effects of frosts and frosts followed by fire on mutualistic interactions between ants and EFN-bearing plants. These disturbances, although common in the Cerrado, are being intensified due to anthropogenic actions and require attention due to their direct interference in ecosystem dynamics. Here, we demonstrate how frosts and fire can affect the richness and diversity of ants and, consequently, the interactions these organisms promote with individuals at other trophic levels. Fire accelerated the regrowth of *O. spectabilis* in a Cerrado previously affected by frosts. With regrowth, EFN-bearing plants attracted ants that efficiently removed herbivores, significantly reducing leaf herbivory rates in this species compared to a species without EFNs or when ant access was prevented through experimental manipulation. Thus, this study suggests that fire plays an important role in restoring ecological interactions in the Cerrado, impacting plants less negatively than frost, even though fire clearly results in a reduction in herbivore fauna. We conclude that disturbances like fire and frost affect species and can have consequences for the dynamics of organisms that provide important ecosystem services. Therefore, this study alerts us that in the face of the rapid changes the planet is undergoing, understanding how such phenomena impact organisms and their interactions is essential to help us prevent and/or mitigate their effects.

## Figures and Tables

**Figure 1 plants-12-03592-f001:**
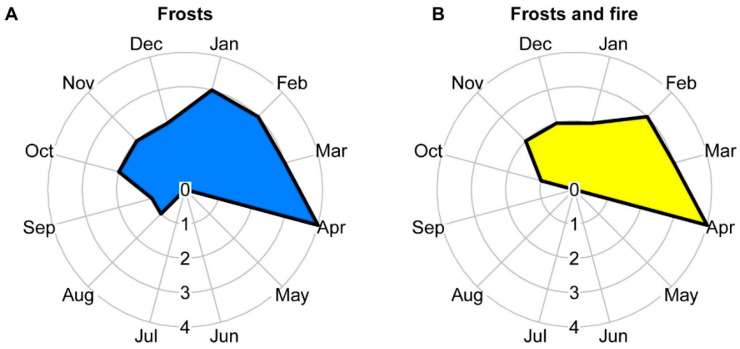
Regrowth of *Ouratea spectabilis* under the effects of frosts that occurred at the end of July (**A**) and under the effects of frosts and subsequent fire that occurred in September (**B**). In the graphs, the intensity of regrowth of individuals is represented from 0 to 4, with 0 = absent, 1 = present from 1 to 25%, 2 = present from 26 to 50%, 3 = present from 51 to 75%, and 4 = present from 76 to 100%.

**Figure 2 plants-12-03592-f002:**
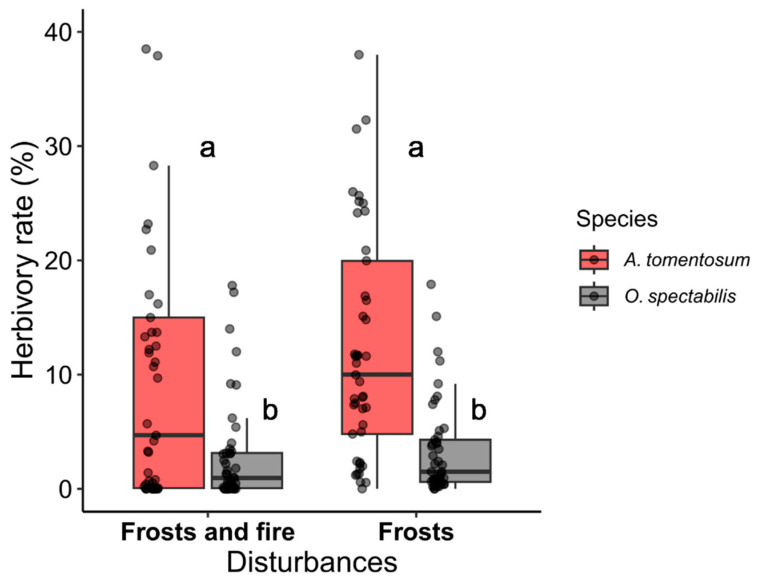
Herbivory rates of plants with EFNs (*Ouratea spectabilis*) and without EFNs (*Aspidosperma tomentosum*) in the area under the effects of frosts and in the area under the effects of frosts and fire. The bars represent median values with interquartile range, maximum, and minimum values, together with raw data points. The different letters at the top of the bars indicate statistically significant differences.

**Figure 3 plants-12-03592-f003:**
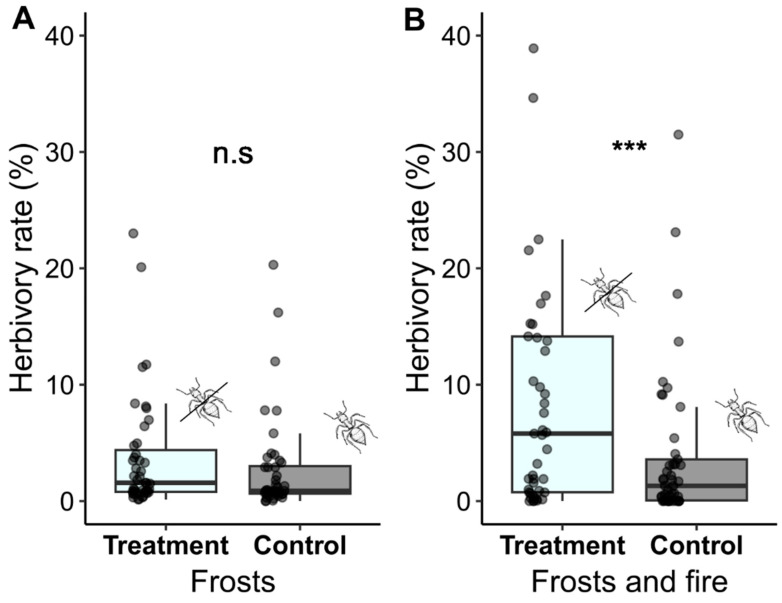
Herbivory compared in *Ouratea spectabilis*, treatment branches (ants excluded from the plants), and control branches (ants with unrestricted access to the plants), under the effects of frosts (**A**) and frosts and fire (**B**). The bars represent median values with interquartile range, maximum, and minimum values, together with raw data points. The symbol *** indicates significant differences, and (n.s) indicates non-significant differences.

**Figure 4 plants-12-03592-f004:**
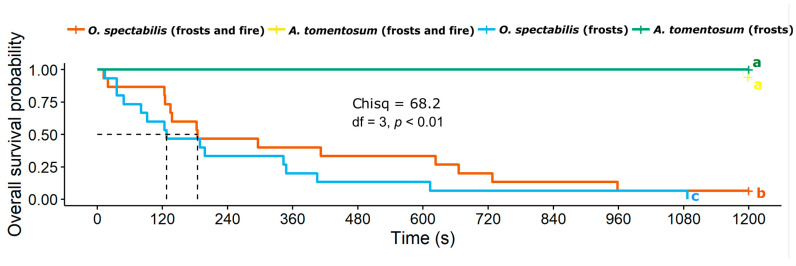
Variation in the survival probability of herbivores (termites) in the presence of ants between a plant species with extrafloral nectaries and a species without extrafloral nectaries. Different letters at the end of the curve indicate significant differences from each other by the paired log-rank test with *p*-value corrected by Benjamini and Hochberg.

**Figure 5 plants-12-03592-f005:**
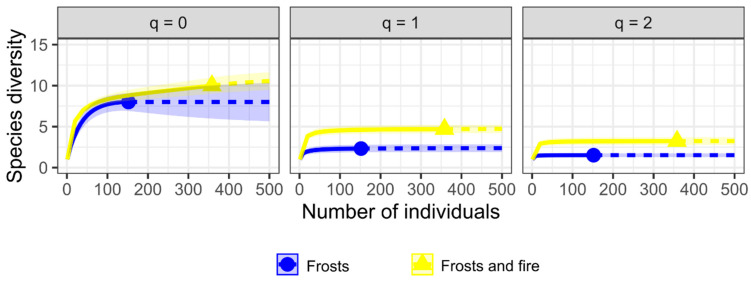
Interpolation with the observed data (solid lines) and extrapolation (dashed lines, extrapolation to 500 individuals) of ant species diversity, on plants of *Ouratea spectabilis*, for two areas under the effects of disturbances caused by frosts and frosts and fire. Parameter q = 0 corresponds to species richness, q = 1 corresponds to diversity using the Shannon index and q = 2 corresponds to species dominance using the Simpson index. Shaded areas represent 95% confidence intervals.

**Figure 6 plants-12-03592-f006:**
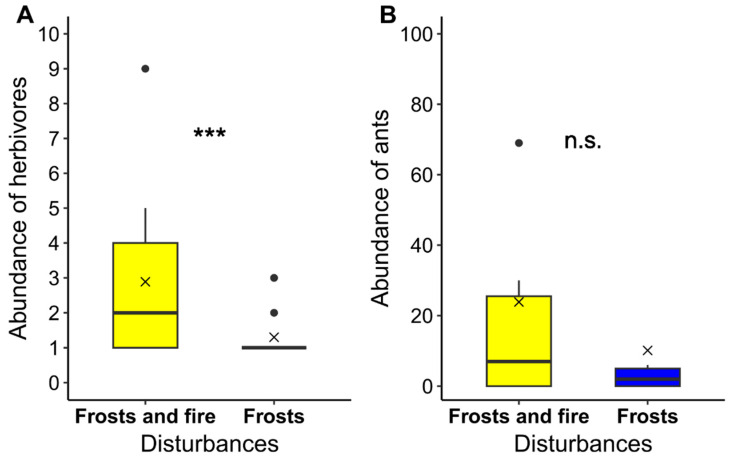
Comparison between the mean abundance of herbivores (**A**) and ants (**B**) on plants of *Ouratea spectabilis* in areas under the effects of frosts and fire and frosts. The GLM indicated significant differences in herbivore abundance between the areas under the different disturbances (*p* = 0.05). The ANOVA did not show significant differences in the mean abundance of ants (*p* > 0.05). The means are represented by the symbol x. The symbols *** and n.s. indicate significant and non-significant differences, respectively.

## Data Availability

The data presented in this study are available on request from the corresponding author. The data are not publicly available due to privacy.

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
