# Peer review of "Extrafloral Nectary-Bearing Plants Recover Ant Association Benefits Faster and More Effectively after Frost-Fire Events Than Frost"

_plants, 2023, doi:10.3390/plants12203592_

Round 1
Reviewer 1 Report
General comments
In the manuscript “Extrafloral nectary bearing plants regrow and recover the benefits of ant association quickly and better after fire than frost” the Authors assess the effects of frosts and frosts followed by fire on the interactions between ants and EFN-bearing plants in the Brazilian tropical savanna, specifically in the Cerrado area.
This is a timely field study that will undoubtedly contribute to building knowledge on the impact of extreme climatic events on ecological interactions in a tropical biome. The paper is well-written and very interesting to read. I have only a few questions/comments regarding the methodology and some suggestions to improve the clarity of the text.
Specific comments
Page 3, line 99: Please rephrase as follows: This highlights the limited knowledge of the impact of this physical stressor on animals and biotic interactions in tropical regions.
Page 4, lines 150-157-158: In the M&M section the Authors use “savannah” while elsewhere in the text they use “savanna”. Please, be consistent with the spelling used.
Page 4, line 174: I suggest indicating in the “Study system” section (and possibly in the introduction) that Aspidosperma tomentosum was chosen as a control for the herbivory rate experiment because it does not possess EFN. This is explicitly stated only later in the text.
Page 5, line 199: The Authors need to provide more details about the sampling methodology employed to characterize the abundance of herbivores and ants. Moreover, I wonder if sampling ant and herbivore guilds in diurnal hours could have influenced the results, given that for instance some ant species are nocturnal.
Page 5, line 227: “However, the plants had to have at least one ant species foraging to initiate the experiment.” Does this mean that the Authors selected only the plants with ants for the experiments? If this is the case, then the individuals were not randomly selected, as indicated in line 220. Please, clarify.
Page 7, line 294: Replace “NEFs” with “EFNs”.
Page 7, line 294: “The average herbivory rate of species with and without NEFs in the area under the effects of frosts were respectively (O. spectabilis, 1.22 ± 0.200 and A. tomentosum, 2.55 ± 0.188; mean ± standard deviation), and in the area under frosts and subsequent fire (O. spectabilis, 1.06 ± 0.203 and A. tomentosum, 2.48 ± 0.188; mean ± standard deviation).”
This sentence does not read well. Please rephrase.
Figure 3, line 328: Please add the letter “a” and “b” to the figure.
Page 9, line 382: It is not clear in the M&M how the samples for the abundance analysis of herbivores and ants were collected.
Page 10, line 410: Please remove “Independently of the presence of EFNs”. Your first hypothesis, as stated in the Introduction, does not test whether plants with EFNs would regrow more quickly compared to plants without EFNs.
Page 12, line 525: Replace “;” with “,”.
Author Response
We thank the reviewer for your kind words, and we are very proud to hear such laudatory comments from a reviewer of Plants.
Page 3, line 99: Please rephrase as follows: This highlights the limited knowledge of the impact of this physical stressor on animals and biotic interactions in tropical regions. - Done
Page 4, lines 150-157-158: In the M&M section the Authors use “savannah” while elsewhere in the text they use “savanna”. Please, be consistent with the spelling used. - Done
Page 4, line 174: I suggest indicating in the “Study system” section (and possibly in the introduction) that Aspidosperma tomentosum was chosen as a control for the herbivory rate experiment because it does not possess EFN. This is explicitly stated only later in the text. -Done
Page 5, line 199: The Authors need to provide more details about the sampling methodology employed to characterize the abundance of herbivores and ants. Moreover, I wonder if sampling ant and herbivore guilds in diurnal hours could have influenced the results, given that for instance some ant species are nocturnal. – Done, Collections were made during the daytime.
Page 5, line 227: “However, the plants had to have at least one ant species foraging to initiate the experiment.” Does this mean that the Authors selected only the plants with ants for the experiments? If this is the case, then the individuals were not randomly selected, as indicated in line 220. Please, clarify. – We fixed it. The intention was to convey that, generally, at least one ant was present on the plants, occasionally. We removed the sentence to avoid any confusion.
Page 7, line 294: Replace “NEFs” with “EFNs”.
Page 7, line 294: “The average herbivory rate of species with and without NEFs in the area under the effects of frosts were respectively (O. spectabilis, 1.22 ± 0.200 and A. tomentosum, 2.55 ± 0.188; mean ± standard deviation), and in the area under frosts and subsequent fire (O. spectabilis, 1.06 ± 0.203 and A. tomentosum, 2.48 ± 0.188; mean ± standard deviation).”
This sentence does not read well. Please rephrase. - Done
Figure 3, line 328: Please add the letter “a” and “b” to the figure. - Done
Page 9, line 382: It is not clear in the M&M how the samples for the abundance analysis of herbivores and ants were collected. – Done, we added a sentence.
Page 10, line 410: Please remove “Independently of the presence of EFNs”. Your first hypothesis, as stated in the Introduction, does not test whether plants with EFNs would regrow more quickly compared to plants without EFNs. – Done, thank you.
Page 12, line 525: Replace “;” with “,”. – Done

Reviewer 2 Report
The paper is very interesting and presents relevant information on the effect of fire and frost disturbances on the presence of NEF and the effect on the ant-plant-herbivore interaction. The author present a good reserch design in order to test the effect of fire and frost in the plant growth and EFNs production in the Cerrado, and their results support that fire restore ecological interactions better than frost, with less impact on plants, ants, and herbivores.
There are some points that must be clarified in the methods, as the variation in trees tall, and also the comparisson in the hervibory rates in both studied species.

Author Response
We thank the reviewer for your kind words, and we are very proud to hear such laudatory comments from a reviewer of Plants.
This reviewer sent us a series of small suggestions and corrections in a PDF version. We accepted all the corrections, and they are highlighted in red in the text. This reviewer suggested a series of references to be included and most of them were accepted.
